# A Central Contribution of TG2 Activity to the Antiproliferative and Pro-Apoptotic Effects of Caffeic Acid in K562 Cells of Human Chronic Myeloid Leukemia

**DOI:** 10.3390/ijms232315004

**Published:** 2022-11-30

**Authors:** Giordana Feriotto, Federico Tagliati, Arianna Brunello, Simone Beninati, Claudio Tabolacci, Carlo Mischiati

**Affiliations:** 1Department of Chemical, Pharmaceutical and Agricultural Sciences, University of Ferrara, 44121 Ferrara, Italy; 2Department of Neuroscience and Rehabilitation, University of Ferrara, 44121 Ferrara, Italy; 3Department of Biology, University of Rome “Tor Vergata”, 00133 Rome, Italy; 4Department of Oncology and Molecular Medicine, Istituto Superiore di Sanità, 00161 Rome, Italy

**Keywords:** Chronic Myeloid Leukemia, transglutaminase type 2, natural molecules

## Abstract

Caffeic acid (CA) has shown antitumor activity in numerous solid and blood cancers. We have recently reported that CA is active in reducing proliferation and triggering apoptosis in both Imatinib-sensitive and resistant Chronic Myeloid Leukemia (CML) cells. Tissue transglutaminase type 2 (TG2) enzyme is involved in cell proliferation and apoptosis of numerous types of cancer. However, its activity has different effects depending on the type of tumor. This work investigated the possible involvement of TG2 activation in the triggering of CA-dependent anticancer effects on the K562 cell line, which was studied as a model of CML. CA-dependent changes in TG2 activity were compared with the effects on cell proliferation and apoptosis. The use of *N*-acetylcysteine (NAC), an antioxidant molecule, suggested that the antiproliferative effect of CA was due to the increase in reactive oxygen species (ROS). The use of a TG2 inhibitor showed that TG2 activity was responsible for the increase in ROS generated by CA and reduced both caspase activation and triggering of CA-dependent apoptosis. The knocking-down of *TGM2* transcripts confirmed the crucial involvement of TG2 activation in CML cell death. In conclusion, the data reported, in addition to ascertaining the important role of TG2 activation in the antiproliferative and pro-apoptotic mechanism of CA allowed us to hypothesize a possible therapeutic utility of the molecules capable of triggering the activation pathways of TG2 in the treatment of CML.

## 1. Introduction

Chronic Myeloid Leukemia (CML) is a myeloproliferative disease caused by a translocation between chromosomes 9 and 22 (called Philadelphia chromosome) resulting in the fusion of two genes, *ABL1* (Ch9) and *BCR* (Ch22). The chimeric product of this fusion (*BCR*-*ABL* oncogene) is a protein with constitutive tyrosine kinase activity leading to uncontrolled proliferation, protection from apoptosis, and promotion of invasion. Although the development of BCR-ABL1 tyrosine kinase inhibitors (TKIs), such as Imatinib, Dasatinib, and Nilotinib, has led to significant achievement in CML therapy, resistance occurs in patients with advanced disease [1].

Tissue transglutaminase (transglutaminase type 2; TG2) is a calcium-dependent enzyme of the transglutaminase family (EC 2.3.2.13). TG2 catalyzes specific post-translational modifications of proteins through a transamidation reaction and is involved in various additional enzymatic activities, such as guanine nucleotide binding and hydrolysis, protein kinase, and disulfide isomerase activities [2]. TG2 belongs to a wide family of proteins involved in various biological mechanisms justified by its interaction with other cellular components, in particular biogenic polyamines [3]. TG2 expression increases in many tumors, especially in those characterized by resistance to chemotherapy or isolated from metastatic sites [2]. Transcription of the *TGM2* gene produces alternative transcripts. The protein TG2 (full-length TG2 protein, 80 kDa) derives from the translation of the *TGM2* transcript *variant 1*. It is synthesized in a closed inactive form but can be switched to the active open form by the intracellular release of Ca^2+^. In addition, 5 other alternative variants have been reported in the literature [4]. TG2 protein is involved in proliferation, apoptosis, and differentiation [5,6]. Both the amount and the activity of TG2 were implicated in cancer cell proliferation. On one hand, the protein level was found to increase in many types of cancer with respect to their normal counterparts [7,8,9,10,11]. On the other hand, TG2 cross-linking activity was inversely related to cell proliferation, as the inhibition of TG2 activity by resveratrol resulted in the normalization of cell proliferation in cholangiocarcinoma cells [12]. In leukemia cells, the administration of All-Trans Retinoic Acid (ATRA) reduced cell proliferation in conjunction with the increase in TG2 activity [13], and in CML patients, the activity of TG2 was lower in tumors than in normal cells [14]. These observations suggested that activation of TG2 could have possible therapeutic applications. TG2 activity, among the other functions, sustains sequential events that led to caspases activation [15]. Furthermore, literature data reported the strong association between oxidative stress and TG2 up-regulation, which in turn may result in cell survival or apoptosis, depending on cell type, kind of stressor, duration of insult, TG2 intracellular localization, and its activity state [16]. The increment of particular *TGM2* transcript isoforms can also modify the intracellular balance of inactive/active TG2. In particular, in leukemia cells, two variants of the *TGM2* transcript have been described: *variant 2*, which encodes the 63 kDa TGH protein and lacks the GTP binding site and therefore always active [17]; *variant 3*-*HEL*, which encodes the 37 kDa TGH2 protein and lacks cysteine in the catalytic core, so it is inactive and cannot be activated [18].

Caffeic acid (CA) belongs to the hydroxycinnamic acid class of plant secondary metabolites and is found in several food sources (coffee, red wine, blueberries, and apples) [19]. This polyphenolic compound has pharmacological activity in some types of human cancer [20]. Our group has recently described the antiproliferative and pro-apoptotic effects of CA in K562 cells, a BCR-ABL positive human CML cell line, on which it also enhanced the antitumor effect of Imatinib [21]. The activation of TG2 obtained through anticancer drugs seems to favor the apoptosis of cancer cells. For example, the cytotoxic drug cisplatin induced apoptosis in a TG2-dependent manner in a model of human hepatocarcinoma [22], and the activation of TG2 increased the frequency of apoptosis in human breast cancer cells [23].

Therefore, in this study, the possible role of TG2 in the proliferation and apoptosis of CML cells, in particular after the treatment with CA, was examined. The results, in addition to shedding light on the antiproliferative/pro-apoptotic mechanism of CA by clarifying whether TG2 is involved, should have possible implications in the treatment of CML.

## 2. Results

### 2.1. The Increase in the Intracellular Oxidative State Was Responsible for the Antiproliferative Effect of CA in CML Cells

The dose-dependent effect of three-day administration of CA (20, 40, 80 µM) to K562 cells, both on the level of reactive oxygen species (ROS) and the cell proliferation, was evaluated. A dose-dependent increase of ROS, with respect to untreated control cells, was evident at CA concentrations above 40 µM (Figure 1A, upper side). At these concentrations, the CA showed a clear antiproliferative effect (Figure 1A, lower side). This observation suggested that the ROS level and the antiproliferative effect produced by CA could be related or concomitant. To verify this hypothesis, 38 µM *N*-acetyl cysteine (NAC) was administered simultaneously for three days with CA (25, 50, 100 µM) and then ROS level and cell proliferation were assayed. As shown in Figure 1B, upper side, NAC effectively inhibited or strongly reduced the CA-induced ROS production at all the concentrations tested. In addition, NAC mitigated the antiproliferative effect of CA (Figure 1B, lower side). These results indicated that CA treatment increased the intracellular oxidative state, which in turn produced the antiproliferative effect.

### 2.2. TG2 Activity Supported the CA-Increased Level of ROS

The role of TG2 activity on CML cell proliferation has not yet been elucidated. In CML patients, TG2 activity present in leukemic cells is lower than in leukocytes of healthy subjects [14], suggesting that dysregulated TG2 activity may play a role in cancer cell proliferation. Furthermore, it has been reported that ATRA exerted its antiproliferative effect on erythroleukemia (HEL) cells through a mechanism that involved increasing TG2 activity [13]. Based on this information, it was assessed whether treatment with CA stimulated TG2 activity in the CML cells (Figure 2). The cells were treated for three days in the presence of CA (40 µM), as single administration or combined with the cell-permeable TG2 inhibitor, R283 (200 µM). Alternatively, as a positive control of TG2 activity, the cells were treated with the calcium ionophore calcimycin (2 µM). On the third day, the TG2 activity (Figure 2A) and the ROS level (Figure 2B) were assessed. The results showed that CA administration increased the activity of TG2 over the normal level present in the untreated cells in a similar way to that obtained with calcimycin while the TG2 inhibitor strongly contained the CA-triggered ROS increment. These results strongly indicated a direct contribution of TG2 activity on increasing the level of ROS with CA treatment.

### 2.3. Effect of CA Treatment on TGM2 Gene Transcription

The increase in TG2 activity following the CA treatment described above could also be due to a concomitant increase in the level of *TGM2* transcripts. In fact, it has been reported in the literature that an increment of intracellular oxidative stress can activate ROS-sensitive transcription factors involved in the regulation of the *TGM2* promoter [4,24]. The *TGM2* gene encodes 6 transcripts generated by differential splicing of the primary messenger. The *TGM2* transcripts expressed in K562 cells were first characterized by the qualitative RT-PCR analysis shown in Figure 3A, which was performed using different pairs of primers able to sort out between the different *TGM2* transcripts [taken from ref. [25] or unpublished primers designed by our laboratory, see Section 4].

The presence of *TGM2* transcript *variants 1*, *2* (tTGH), and *3*-*HEL* (tTGH2) were identified, while *variants 3*, *4*, and *5* were absent or not detectable by RT-PCR. Therefore, the expression of *TGM2* transcripts was assayed by real-time RT-qPCR (Figure 3B). The transcript *variant 1* was 93.25% of the total *TGM2* mRNA and *variant 2* was 6.7%, while *variant 3HEL* was scarcely expressed, representing only 0.05%. Afterward, it was verified whether a three-day treatment with CA (38 µM) modified the level of expression of *TGM2* transcripts *variant 1* and *2* by RT-qPCR (Figure 3C). The results showed no appreciable differences in the expression levels of each *TGM2* transcript before and after the treatment. This allowed us to deduce that treatment with CA increased the TG2 activity only by triggering post-translational mechanisms rather than by modifying transcription levels of the *TGM2* gene.

### 2.4. Contribution of TG2 on CA-Induced ROS Level Increase

The effect of TG2 protein reduction on the oxidative state was evaluated in CML cells. For this purpose, starting from the K562 cell line, a modified cell line (KD-TGM2) was generated. This cell line was obtained by stably introducing an inducible expression cassette into the cell genome, which allows the expression of a small interfering RNA directed against a common region present in all known *TGM2* gene transcripts (the region 657–1101 of the classical *TGM2* transcript, *variant 1*, see Figure 4A). In these cells, the addition of doxycycline (DOX) reduced the level of *TGM2* transcripts (Figure 4D), and therefore of TG2 protein (Figure 4C), by half. In the first set-up experiments, the KD-TGM2 cells were grown in the presence of different DOX concentrations to determine the maximum concentration of DOX that can be administered to the cells without causing an appreciable cytotoxic effect (Figure 4B). In the following set-up experiments, DOX concentration was set to 4 µg/mL to induce the expression of shRNA-*TGM2* without appreciable effects of cell proliferation. To determine the time length useful to reduce the *TGM2* transcripts, the expression levels of the TG2 protein were assayed 24 and 48 h after the DOX addition (Figure 4C). The results showed a considerable reduction in TG2 protein amount of 48% and 55%, respectively, after 24 and 48 h of induction compared to non-induced control cells. The reduction of *TGM2* transcripts was also confirmed through RT-qPCR analysis performed after 48 h of induction (Figure 4D). For this purpose, the primer pair TGM2-ALL (see Figure 4A) was used, which amplified all the known *TGM2* transcripts. As anticipated above, the expression of the *TGM2* transcript was reduced by more than 50% after 48 h, hence this time length was used.

Finally, the contribution of TG2 on CA-induced ROS level was evaluated. To this end, increasing concentrations of CA (0–80 µM) were administered to KD-TGM2 cells expressing normal or reduced levels of TGM2, and the ROS levels were analyzed 3 days later (Figure 4E). In cells with reduced *TGM2* transcripts level (+DOX), the data evidenced a smaller increase in ROS level after CA administration, with respect to cells expressing normal TG2 level (−DOX). Since, as reported above, the CA-treatment itself did not affect *TGM2* transcription (see Figure 3C), taken together, this data suggested that the activation of TG2 rather than an increase in its expression played a fundamental role in determining the increase in the intracellular oxidative state following treatment with CA.

### 2.5. TG2 Activity Sustained the CA-Triggered Apoptotic Mechanism

In previous work, our research group showed how the antiproliferative activity of CA is due to the ability of this molecule to open the transition pore of mitochondrial permeability [21]. This event causes the depolarization of the mitochondrial membrane potential, which triggers a caspase-dependent apoptosis mechanism, with consequent exposure of phosphatidylserine on the outer side of the cell membrane [21]. In addition, it was reported in the literature that TG2 activates apoptosis through a caspase-dependent or caspase-independent mechanism [15].

Therefore, herein, it was evaluated whether TG2 activity was part of the caspase-dependent apoptosis mechanism triggered by CA (Figure 5). The K562 cells were treated with CA (38 µM) in the presence or absence of the inhibitor R283 (200 µM). Then, they were analyzed after 24 h for caspase activity via z-VAD-FITC staining (Figure 5A) or after 72 h for the exposure of phosphatidylserine via staining with Annexin V-FITC (Figure 5B) and subsequent flow cytometric analysis. The results shown in Figure 5A highlighted that caspase’s activation, obtained with CA treatment, decreased in the presence of the TG2 inhibitor, and the decrease was significant (*n* = 6, *p* < 0.05). This result represented a further confirmation that the activation of caspase activity occurs through a TG2-dependent mechanism. The results presented in Figure 5B showed that inhibition of TG2 activity in the presence of R283 reduced the percentage of apoptotic cells, corroborating the role of TG2 in triggering apoptosis in CML cells. Therefore, the activity of TG2 participates both in caspase activation and in the apoptotic mechanism triggered by CA.

## 3. Discussion

Phenolic acids of plant origin are gaining growing interest in the medical field and, in particular, have shown interesting anticancer activities [26]. CA is not just a potential anticancer molecule per se, but it can also potentiate the effect of current anticancer drugs by delaying in vivo cancer progression. In fact, the administration of CA increased the anticancer effect of paclitaxel in nude mice inoculated with H1299 lung cancer cells [27]. These premises attracted our interest in this natural molecule.

Recently our research group has shown that CA is also active in CML, in which it is able to slow down cell proliferation and trigger apoptosis in both sensitive and resistant cells to Imatinib, which is the first-line drug in the therapy of this type of tumor [21]. Studies conducted on rats fed a diet rich in CA have shown that this natural molecule is absorbed through the intestinal wall and enters the bloodstream, in which it reaches high concentrations [28,29,30] of the same magnitude as those capable of triggering apoptosis in K562 cells [21].

However, after intestinal absorption, various enzymes such as sulfotransferase, UDP-glucosyltransferase, and catechol-o-methyltransferase, which catalyze methylation, sulfation, and glucuronidation reactions, modify the CA molecule [31] and, therefore, also its anti-leukemic potential in vivo. On one hand, the chemical synthesis of caffeic acid derivatives could be the way to obtain novel molecules with better biological stability and, possibly, greater antitumor effects. On the other hand, knowledge of the antitumor mechanism triggered by CA could suggest ways to enhance the biological activity of the natural molecule. Starting from this assumption, evidence on the mechanisms underlying the anti-leukemic effect of CA is described in the present study.

It is well known that the caspases cascade activates the intrinsic pathway of apoptosis causing DNA damage. In this respect, in melanoma cells, administration of CA induced apoptosis and cell cycle arrest through the activation of the caspase pathway [32]. It has been hypothesized in other tumors that CA triggered the mechanism by increasing the intracellular oxidative state [33]. According to the literature, the results herein presented showed that, even in CML cells, CA determined marked cytotoxicity accompanied by caspases activation and increased dose-dependent ROS production. The key role of ROS production in the cytotoxic mechanism of CA was demonstrated by the use of NAC, which allowed the recovery of proliferation.

TG2 is a protein that modulates numerous biological mechanisms including cell proliferation and apoptosis [4]. TG2 expression was found largely increased in many tumor types, especially in those characterized by resistance to chemotherapy or isolated from metastatic sites [4]. The inhibition of TG2 activity restored cell proliferation of G1/S-blocked cholangiocarcinoma cells following treatment with resveratrol. This led to the conclusion that TG2 has involved in tumor cell proliferation [12]. Instead, the role of TG2 activity in CML was not yet elucidated and little evidences suggested its possible role in leukocyte proliferation. First, TG2 activity in leukemic cells of CML patients has been observed to be lower than that found in leukocytes of healthy subjects [14]. Second, it has been reported that the increase in TG2 activity, raised after ATRA treatment, reduced the proliferation of leukemia cells [13]. Even if the literature reported that oxidative stress can modify TG2 activity by acting through transcriptional mechanisms involving ROS-sensitive transcription factors [4], data showed in Figure 2A highlighted that, at least in CML cells, also the TG2 activity contributed to increasing the intracellular oxidative stress. From the data reported in Figure 2B, the use of the TG2 inhibitor blocked the increase in ROS level following treatment with CA, suggesting that TG2 activity was a fundamental requirement to determine the raising of ROS. This observation was confirmed through the knock-down of the *TGM2* transcripts (see Figure 4E), which showed that the lowering of TG2 expression strongly reduced the level of ROS achieved after treatment with CA. Therefore, a tight link emerged between TG2 activity, oxidative state, and antiproliferative effects following treatment with CA. In particular, the data suggested that the CA determined its antiproliferative effect by activating TG2, and this, in turn, mediated the augment of the ROS level.

Although an increase in TG2 activity can be achieved with the upregulation of the *TGM2* transcript *variant 2*, which produces the constitutively active tTGH isoform, treatment with CA did not change the expression of this transcript. Therefore, the increase in intracellular levels of active TG2 should probably be due to the conversion of inactive TG2 already present in leukemic cells into its active form. We did not investigate which mechanism activated TG2, but it is reasonable to assume that it occurred through increased intracellular Ca^2+^ levels triggered by CA treatment. In this respect, it has been reported that CA induced Ca^2+^ release from the endoplasmic reticulum and Ca^2+^ influx in human gastric cells [34], and CA-derivatives did the same in lymphocytes isolated from peripheral blood [35] and in acute T-cell leukemia cells [36]. These observations suggest that TG2 activation may not only be a mechanism restricted to CML cells but could be a general mechanism triggered by CA.

It has been described in the literature that TG2 can activate apoptosis through a caspase-dependent or caspase-independent mechanism [15]. In CML cells, CA directs CML cells towards apoptosis through the caspase cascade with a mechanism that required the presence of TG2 in an active form since both caspases activation and extroversion of the phosphatidylserine were strongly reduced in the presence of the TG2 inhibitor (Figure 5).

In conclusion, the results summarized in Figure 6 strongly suggested that the activation of TG2 played a central role in the cytotoxicity of CA and the triggering of the pro-apoptotic mechanism in CML cells. Therefore, these findings suggest that TG2 activators could find a possible use in the treatment of CML.

## 4. Materials and Methods

### 4.1. Materials

CA and calcimycin and stock solutions were prepared in dimethyl sulfoxide (DMSO) and stored in the dark at −20 °C in a sealed glass vial until their use. The phosphate-buffered saline (PBS), RPMI-1640 medium, fetal bovine serum (FBS), pen-strep solution, doxycycline, *N*,*N*′-dimethylcasein, 5-(biotinamido)pentylamine, and *N*-acetyl cysteine (NAC) were purchased from Sigma Aldrich (Milan, Italy) while the TG2 inhibitor 1,3,dimethyl-2-[(2-oxopropyl) thio] imidazolium chloride (R283) was a kindly gift from Prof. Martin Griffin (Aston University, Aston Triangle, Birmingham, UK).

### 4.2. Cell Culture and Viability Test

The K562 cell line was cultured in RPMI-1640 medium supplemented with 10% FBS, 100 U/mL penicillin, and 100 μg/mL streptomycin at 37 °C in a humidified atmosphere of 5% CO_2_/air.

Cell proliferation was evaluated with an MTT assay. Cells in the log phase of growth were seeded at 25 × 10^3^/mL into 96-well plates in a fresh medium containing the examined compounds at various concentrations. After 72 h of growth, 100 μL of 0.5 mg/mL 3-(4,5-dimethylthiazol-2-yl)-2,5-diphenyl tetrazolium bromide (MTT) solution was added in the culture medium and the cells were incubated for 4 h at 37 °C. Then, 100 μL of DMSO were added to each well to allow the formed formazan crystals to dissolve. The cells’ viability was assessed with the multi-well plate reader Infinite 200 PRO Tecan (TECAN, Mannedorf, Switzerland) measuring the absorbance at 570 nm. The proliferation in each sample was expressed as percent with respect to untreated control cells. Percentages were graphed as a function of molecule concentration and concentration required for 50% inhibition of the cellular proliferation, the half maximal inhibitory concentration (IC50), was calculated. Three independent experiments were performed in triplicate.

### 4.3. RNA Isolation and RT-PCR

Cells were lysed in 1 mL of TRIzol reagent (Invitrogen, Thermo Fisher Scientific, Milan, Italy) and 200 µL of chloroform was added. The mixture was vigorously shaken, incubated at room temperature for 10–15 min, and centrifuged at 12,000× *g* for 15 min. The aqueous phase was collected and the RNA was precipitated with a isopropanol addition. The pellet washed in 75% ethanol was dissolved in 10 mM Tris-HCl pH 7.5, 1 mM EDTA (ethylenediaminetetraacetic acid). The RNA samples were processed with the DNA-free™ DNA Removal Kit (Invitrogen, Thermo Fisher Scientific) to remove contaminating DNA.

The RNAs (2 µg) were reverse transcribed (RT) with or without the ImProm-II™ Reverse Transcription System (Promega Italia, Milan, Italy) using oligo-dT primers in a standard 20 µL reaction.

In semi-quantitative RT-PCR analysis, PCR reactions were performed using 1 µL of the cDNA reaction mixture as template, 10 pmoles of gene-specific primers, 33 µM dNTPs, and 2 U of Taq polymerase. The *TGM2* transcript variants were amplified using the primer couples described in [25] annealing the exon11 and 12 (TGM2-v1), exon10 and 10b (TGM2-v2), and exon5 and 6b (TGM2-v3HEL). The primers couple, TGM2v1-2,3,4 (F: 5′-ATA AGT TAG CGC CGC TCT CC-3′ and R: 5′-GTG AGG ACA TAC TCC TGC CG-3′), generating transcript-specific bands of different weight size, was used to amplify the *TGM2 variants 1* and *2* (585 bp), *variant 3* (342 bp), and *variant 4* (405 bp). The *TGM2* transcripts *variant 5* were amplified using primers F: 5′- GCACAGGAGACCAAGAGACC-3′ and R: 5′-GTG AGG ACA TAC TCC TGC CG-3′.

In real-time RT-qPCR, 1 µL of the cDNA reaction mixture was used as a template in gene-specific amplification performed on StepOnePlus Real-Time PCR System using the StepOne software v2.3 (Thermo Fisher Scientific, Milan, Italy). The amplifications were performed as previously described [21]. The PCR amplifications were performed in a 50 µL volume containing 25 µL SYBR green PCR master mix (Thermo Fisher Scientific) containing the ROX internal passive reference dye, 0.5 µM of each primer, and optimized MgCl_2_ concentration between 1.5–3 mM. All determinations were performed in triplicate wells. Samples in which the RNA was not retro-transcribed have given CT values comparable to those obtained in the no template control well. Endpoint amplified products were subjected to melt curve analysis to exclude possible variations in the length of the amplified fragments. The relative quantity of target transcript in the sample was calculated with respect to the reference β-actin mRNA using a comparative CT (∆∆CT) method. The relative value was expressed as 2^−∆∆CT^. The absolute quantification of the different *TGM2* transcripts was performed with the interpolation/extrapolation of the standard curve.

### 4.4. TG2 Activity Assay

The cell lysates were prepared by suspending freshly prepared cell pellets with lysis solution (1% Triton-X-100 in 50 mmol/L Tris-HC1 pH 7.5, 150 mmol/L NaCl, 5 mmol/L EDTA 25 mmol/L dithiotreitol) at 4 °C. The mixtures were directly centrifuged at 23,000× *g* for 20 min at 4 °C [14]. The transglutaminase activity was measured essentially using the solid-phase microtiter plate assay described by Slaughter et al. [37]. The polystyrene microtiter plates were coated with *N*,*N*′-dimethylcasein. The substrate 5-(biotinamido)pentylamine was covalently incorporated into *N*,*N*′-dimethylcasein with the transglutaminase activity present in cell lysates. The biotinylated product captured on the well bottom was evidenced by streptavidin-alkaline phosphatase and quantitated by measuring the absorbance at 405 nm following the addition of *p*-nitrophenyl phosphate. As a positive control of TG2 activation, 2 µM calcimycin was administered to the cells. Each experiment was repeated three times in duplicate.

### 4.5. KD-TGM2 Cells Preparation

The shRNA-TGM2 (5′-ATG GTC AAC TGC AAC GAT GA-3′, nucleotides 909–929 of TGM2 transcript variant 1, seq. ID: NM_004613.3) was drawn on the region 657–1101 common to all variants of *TGM2* described in GenBank database. The sequence was inserted into the pSingle-tTS-shRNA Vector (Clontech laboratories Inc., Mountain View, CA, USA) for doxycycline-inducible expression of the shRNA. The plasmid was transfected into K562 cells using the Lipofectamine transfection reagent (Invitrogen, Thermo Fisher Scientific, Milan, Italy), and cells resistant to G418 were selected. Following the addition of doxycycline (DOX) in the medium (3 µg/mL), the expression level of all the *TGM2* variant transcripts was evaluated with RT-qPCR using the primer couple TGM2-ALL (F: 5′-GAC TCG GAA GAG GAG CGG CA-3′; R: 5′-CTC AGC ACT GTG CAG GCC AC-3′) that annealed to all the *TGM2* transcript variants described so far.

### 4.6. Apoptosis Measurement

The cells cultured in the absence or in the presence of each compound for 72 h were dual stained with the Annexin V and Propidium Iodide (PI) and analysed using flow cytometry. Briefly, the cultured cells were collected with centrifugation, then were washed in PBS and fixed in 70% ethanol at −20 °C for 16 h. The cancer cells were incubated in 6-well plates in a medium containing the examined compound for 72 h. Afterward, the control cells growth in a normal medium reached 70–80% of confluence. The cells were harvested using trypsinization and together with the cells floated in the medium were stained using an Annexin V Kit (MabTag GmbH, Friesoythe, Germany) according to the manufacturer’s protocol and immediately analyzed with flow cytometry BD FACS Aria II (Becton Dickinson, Franklin Lakes, NJ, USA). Early apoptotic cells with exposed phosphatidylserine but intact cell membranes bound only to Annexin V-FITC. The cells in late apoptotic stages were labeled with both Annexin V-FITC and PI, whereas necrotic cells were labelled with PI only. Each experiment was repeated three times in duplicate.

### 4.7. Caspase Activity Assay

At the day of analysis, 1 × 10^5^ K562 cells from each experimental point were collected and suspended in 1 mL of RPMI-1640 + 10% FBS containing CaspACE™ FITC-VAD-FMK In Situ Marker (Promega Italia, Milan, Italy) at a final concentration of 10 µM and incubated for 20 min at 37 °C, 5% CO_2_. After this period, cells were centrifuged at 300× *g* for 5 min, washed with PBS, and suspended in PBS to 1 × 10^6^ cells/mL. The cellular fluorescence was acquired with flow cytometry as described above. Each experiment was repeated three times in duplicate.

### 4.8. Assessment of Intracellular Oxygen-Derived Free Radicals

The DCFDA-Cellular ROS Assay Kit/Reactive Oxygen Species Assay Kit (ABCAM, Cambridge, UK) was used to determine the intracellular oxidative stress. The cell-permeant fluorogenic dye 2′,7′-dichlorofluorescin diacetate (DCFDA) measures hydroxyl, peroxyl, and other ROS within the cell. Briefly, the cells were plated on 12 well plates at a concentration of 3 × 10^4^ cells/mL in 2 mL final volume of 10% serum-containing RMPI-1640 medium and incubated with or without treatments. After three days, 1 mL of culture was centrifuged 500× *g* for 1 min and suspended in 200 µL of 1 µM DCFDA. The samples were incubated for 30 min at 37 °C in the dark. Fluorescence was measured on a microplate reader with excitation at 485 nm and emission at 530 nm and expressed as fold with respect to the ROS level present at time zero in the untreated control cells. Each experiment was repeated at least three times in triplicate.

### 4.9. Statistical Analysis

The results were expressed as the arithmetic mean ± standard deviation. Statistical calculations were performed using a one-way ANOVA, and the differences among groups were examined using the Bonferroni *t*-test. The *p* values < 0.05 were considered significant.

## Figures and Tables

**Figure 1 ijms-23-15004-f001:**
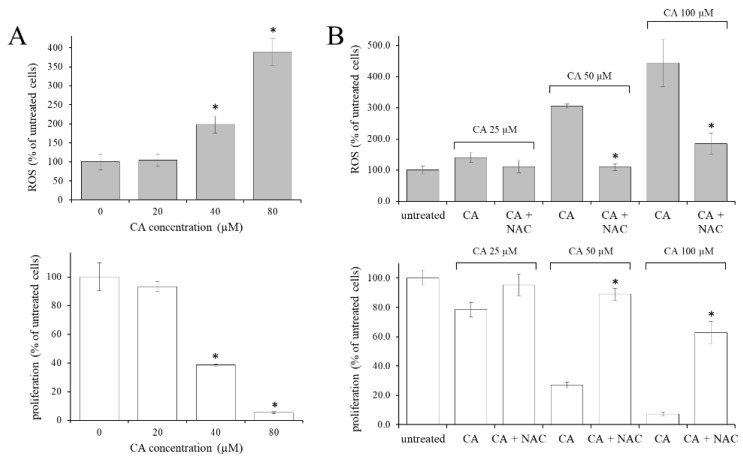
Effect of CA treatment on cell proliferation and on ROS level in K562 cells. (**A**) Dose-dependent effect of CA on ROS level (**upper side**) and cells proliferation (**lower side**). Asterisks indicate significant differences (*p* < 0.05) observed in CA treated cells versus untreated control (0). (**B**) Effects of NAC (38 µM) and CA co-administration on ROS level (**upper side**) and cells proliferation (**lower side**). NAC = *N*-acetyl cysteine, CA = caffeic acid. Asterisks indicate significant differences (*p* < 0.05) obtained upon simultaneous administration of CA and NAC compared to CA-treated cells.

**Figure 2 ijms-23-15004-f002:**
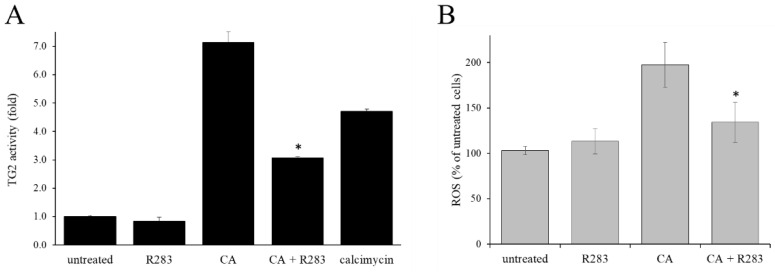
Involvement of TG2 activity in ROS production induced by CA administration in CML K562 cell line. (**A**) Treatment with CA (40 µM) increases the activity of TG2 and the R283 inhibitor reduces this increase. (**B**) The increase in the level of ROS induced by CA is significantly reduced in the presence of the TG2 inhibitor R283 (200 µM). Asterisks indicate significant differences (*p* < 0.05) obtained with the simultaneous administration of CA and R283 compared to CA-treated cells.

**Figure 3 ijms-23-15004-f003:**
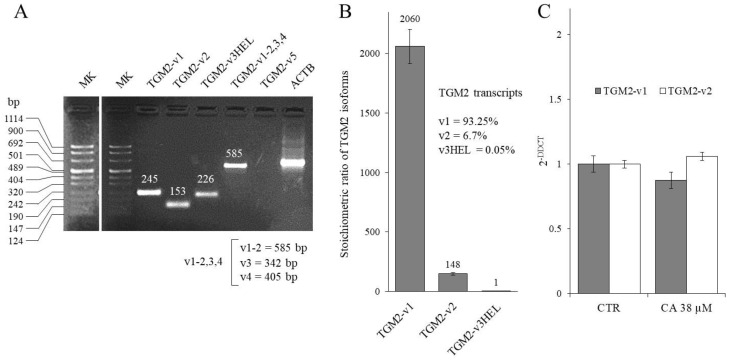
Characterization of *TGM2* variants in K562 cells, before and after the CA-treatment. (**A**) Semi-quantitative RT-PCR analysis of *TGM2* variants. The molecular weights expected from the amplification of the different *TG2M* variants with the primers couple TGM2 v1-2,3,4 are indicated in the lower part of the panel. (**B**) RT-qPCR of the *TGM2* transcript variants expressed in K562 cells. Data are expressed as arithmetic mean ± SD. (**C**) RT-qPCR analysis of predominant *TGM2* transcript variants after the treatment with CA. Data are expressed as arithmetic mean ± SD.

**Figure 4 ijms-23-15004-f004:**
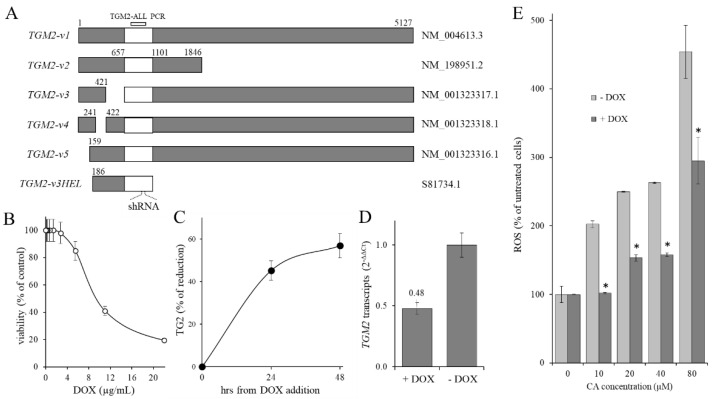
Effect of *TGM2* transcripts knock-down on ROS production. (**A**) BLAST alignment of the known *TGM2* transcript variants. The homology region 657–1101 (numbers refer to TGM2 transcript variant 1) common to all *TGM2* transcript variants was represented as a white box. (**B**) Dose-dependent cytotoxic effect of DOX on KD-TGM2 cells. Viability was measured by MTT assay. 95% of cells were viable by using DOX at 4 µg/mL. DOX = doxycycline. Data are expressed as arithmetic mean ± SD. (**C**) Flow cytometric analysis of time-dependent expression of TG2 protein in KD-TGM2 cells after the addition of DOX 4 µg/mL. Data are expressed as arithmetic mean ± SD (**D**) RT-qPCR analysis of all variants of *TGM2* transcripts expression measured after 48 h of induction. The PCR was performed using the TGM2-ALL primers. Data are expressed as arithmetic mean ± SD. (**E**) Dose-dependent effect of CA administration on *TGM2* transcripts level in KD-TGM2 cells not induced or induced to express the shRNA targeting *TGM2* transcripts. *TGM2* transcripts expression was measured after 48 h of induction by the primer couple TGM2-ALL. Data are expressed as arithmetic mean ± SD. Asterisks indicate significant differences (*p* < 0.05) produced by CA treatment on cells in which knockdown of TG2 (+DOX) transcripts was induced compared to cells with normal levels of transcripts (−DOX).

**Figure 5 ijms-23-15004-f005:**
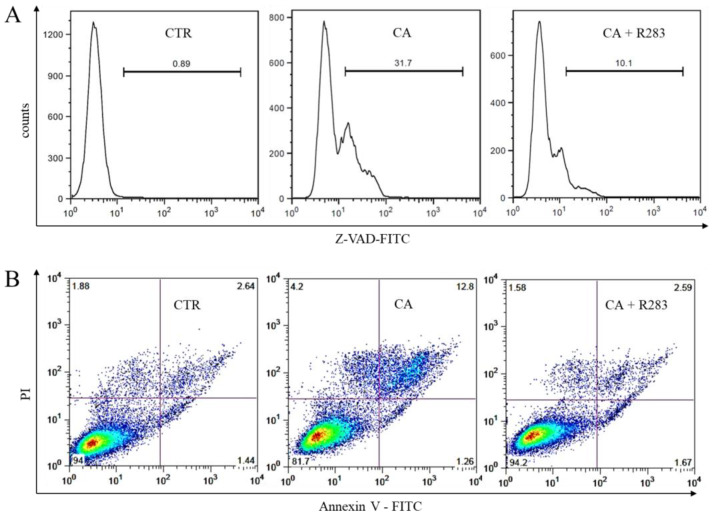
TG2-dependent CA-induced apoptosis in K562 cells. (**A**) The CA activates the caspase cascade 24 h after the start of treatment through a TG2-dependent mechanism. (**B**) The modulation of TG2 activity in the presence of R283 reduces the number of cells in apoptosis measured 72 h from the start of treatment. Representative acquisitions are presented.

**Figure 6 ijms-23-15004-f006:**
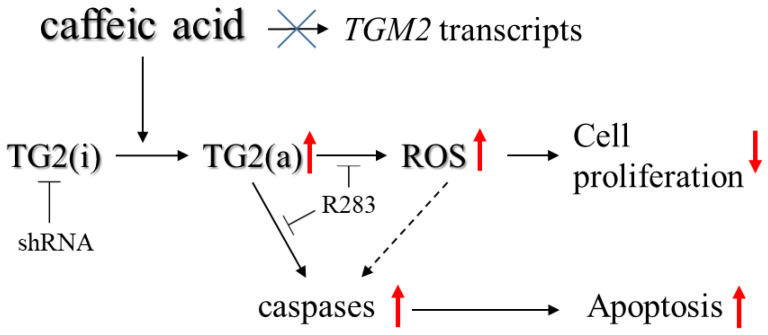
Schematic summary of the effects of CA on cell proliferation and apoptosis in CML cells. TG2 (i) = TG2 inactive; TG2 (a) = TG2 active. Red arrows: modulatory effect of CA-treatment. Treatment with CA does not modify the transcription of the TGM2 gene but increases the level of active TG2 by triggering the conversion of the inactive form into the active one. TG2 activity increases the intracellular oxidative state, which in turn modifies cell proliferation by triggering the caspase pathway leading to cell death by apoptosis.

## Data Availability

Not applicable.

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
