# Peer review of "A Central Contribution of TG2 Activity to the Antiproliferative and Pro-Apoptotic Effects of Caffeic Acid in K562 Cells of Human Chronic Myeloid Leukemia"

_ijms, 2022, doi:10.3390/ijms232315004_

Round 1
Reviewer 1 Report
The author here reported that CA-dependent changes in TG2 activity of K562 cell proliferation and apoptosis. The use of N-acetylcysteine (NAC) demonstrated that the antiproliferative effect of CA was due to the increase in ROS. The use of a TG2 inhibitor showed that TG2 activity was responsible for the increase in ROS generated by CA, and reduced both caspase activation and triggering of CA-dependent apoptosis. The knocking-down of TGM2 transcripts by shRNA confirmed the crucial involvement of TG2 activation in K562 cell death. Authors concluded that in addition to ascertaining the important role of TG2 activation in the antiproliferative and pro-apoptotic mechanism of CA, the molecules capable of triggering the activation pathways of TG2 in the treatment of CML will be possible. The manuscript is interesting and the design is sound reasonable. Before publication, some concerns should be addressed.
Major points:
1. The authors’ data were totally from in vitro experiment. If they did not conduct in vivo research, some results from CML patients’ sample will make the conclusion more reliable instead of only from K562 cell line.
Minor points:
2. In Fig.1, the author failed to give more information about the NAC. For instance, which this inhibitor was from? Besides, how to measure the cell proliferation in their system?
3. Another compound named R283, which was used by the author, the information was not enough again.
4. Author should provide the Statistics tools and analysis. Consequently, related mark in the figures must be added.
5. In addition to the ROS level in Fig.4, please provide the direct shRNA inhibition data on the TGM2 mRNA level.
6. In Fig.5, a quantitative analysis should be performed in addition to the flow spot pictures.
Author Response
Reviewer #1
The authors thank you for the criticisms requested by the reviewer, aimed at improving the quality of this manuscript, which they welcome, trying to respond as far as possible.
Major points:
- The authors’ data were totally from in vitro experiments. If they did not conduct in vivo research, some results from CML patients’ samples will make the conclusion more reliable instead of only from the K562 cell line.
Reply: authors thank the reviewer for this correct observation, which is very important, especially in anticipation of the future use of TG2 activators in the therapy of CML. At this moment, our research group does not have any patients available for the suggested investigation, which will be the subject of a future line of research. In this work, we purposely conducted only in vitro assays since (1) the K562 cell line is a good human cell model of CML, as it was isolated by Lozzio and Lozzio from a CML patient in terminal blast crisis (Lozzio CB, Lozzio BB Human chronic myelogenous leukemia cell-line with positive Philadelphia chromosome. Blood. 1975; 45:321-334), and (2) it is an established cell-line thus providing a definite advantage in terms of data reproducibility over isolated CML cells from patients whose genomic structures vary markedly during this phase of the disease. It would be necessary to set up a precise staging of the patients before being able to carry out the study proposed by the reviewer, and at this moment, we do not have the series available. However, we fully agree with the reviewer and we are involving a hospital clinical group working on CML in our research, but the times to start this study are unfortunately long and tied to the necessary bureaucracy, having to pass the scrutiny of the hospital ethics committee first and the university one then. Precisely because of this impossibility to carry out the experiments requested by the reviewer in a short time, we have always tried to use possible terms to indicate a future interest of the activators in the therapy. However, we believe that the relevance of the biological data described in this work may in any case deserve publication in this form, even if it can be improved. Since data relating to patient CML are missing, we specified in the title that the work was carried out solely on the K562 cell line model of CML. Therefore, the title of the manuscript was changed to “A central contribution of TG2 activity to the antiproliferative and pro-apoptotic effects of caffeic acid in K562 cells of human chronic myeloid leukemia”.
Minor points:
1- In Fig.1, the author failed to give more information about the NAC. For instance, which this inhibitor was from? Besides, how to measure the cell proliferation in their system?
Reply: to answer the first point, the provenance of the NAC has been added in section “4.1 Materials”. As regards cell proliferation, we added in the section “4.2. Cell culture and viability test” the sentence “Cell proliferation was evaluated by MTT assay”.
2 - Another compound named R283, which was used by the author, the information was not enough again.
Reply: the sentence “while the TG2 inhibitor 1,3,dimethyl-2-[(2-oxopropyl) thio] imidazolium chloride (R283) was a kind gift of Prof. Martin Griffin (Aston University, Aston Triangle, Birmingham, UK.)” has been added in section “4.1 Materials”.
3 - Author should provide the Statistics tools and analysis. Consequently, related mark in the figures must be added.
Reply: The paragraph "4.9 Statistical analysis" has been added in the Materials and Methods section. In the caption of each figure, asterisks have been added to underline the significant values and specified that the value reported in the figure is the arithmetic mean ± SD.
4 - In addition to the ROS level in Fig.4, please provide the direct shRNA inhibition data on the TGM2 mRNA level.
Reply: the requested data were previously presented in Fig.4D
5 - In Fig.5, a quantitative analysis should be performed in addition to the flow spot pictures.
Reply: In figure 5 we reported the flow cytometric plot which represents a representative plot of three independent experiments conducted in duplicate. However, in the text on pg.7, the sentence was transformed by “The results shown in Fig.5A, highlighted that caspase’s activation obtained by CA treatment, decreased in the presence of the TG2 inhibitor”, in “The results shown in Fig.5A, highlighted that caspase’s activation, obtained by CA treatment, decreased in the presence of the TG2 inhibitor and the decrease was significant (N=6, p<.05).”
Reviewer 2 Report
I applaud the authors for the extensive literature search and background. The importance of CA and TG2 role has been well described and shown here that there was a CA-dependent changes in TG2 activity and ascertaining the role of tg2 activation in antiproliferative and pro-apoptotic mechanism of CA.. Overall, the structure and logic of the manuscript is clear and well flowing. However, I found there was no statistical analysis performed on the datasets. It could be important to run those tests and support the data with any significant results.
Line 44: What are the reasons for resistance? Is it just advanced disease state?
Line 118: Define ATRA
Figures:: Were there any statistical tests that were run on these data sets? If so, please indicate any significant bars
Author Response
Reviewer #2
The authors thank you for the criticisms requested by the reviewer, aimed at improving the quality of this manuscript, which they welcome, trying to respond as far as possible.
MINOR:
1- Line 44: What are the reasons for resistance? Is it just advanced disease state?
Reply: we did not understand exactly which part of the text the request for clarification refers to. We believe it may refer to the phrase “TG2 expression increases in many tumors, especially in those characterized by resistance to chemotherapy or isolated from metastatic sites [2].“ In other tumors, it has been shown that the conformation in which TG2 is found it can influence the proliferation (closed structure, not activated, through the GTPase domain) and therefore that an unregulated mechanism of activation of it can be a mechanism of escape from the pharmacological treatment. Also, in the open (active) conformation TG2 showed the opposite effect, pro-apoptotic, and for this reason, we wanted to clarify the role of its activity in CML.
2 - Line 118: Define ATRA
Reply: ATRA description was added in the first place of appearance (line 70)
3 - Figures:: Were there any statistical tests that were run on these data sets? If so, please indicate any significant bars
Reply: The paragraph "4.9 Statistical analysis" has been added in the Materials and Methods section. In the caption of each figure, asterisks have been added to underline the significant values and specified that the value reported in the figure is the arithmetic mean ± SD.